Resource

# Sequence and expression levels of circular RNAs in progenitor cell types during mouse corticogenesis

Martina Dori[1] , Leila Haj Abdullah Alieh[1], Daniel Cavalli[1] , Simone Massalini[1], Mathias Lesche[2] , Andreas Dahl[2] , Federico Calegari[1]

Circular (circ) RNAs have recently emerged as a novel class of transcripts whose identification and function remain elusive. Among many tissues and species, the mammalian brain is the organ in which circRNAs are more abundant and first evidence of their functional significance started to emerge. Yet, even within this well-studied organ, annotation of circRNAs remains fragmentary, their sequence is unknown, and their expression in specific cell types was never investigated. Overcoming these limitations, here we provide the first comprehensive identification of circRNAs and assessment of their expression patterns in proliferating neural stem cells, neurogenic progenitors, and newborn neurons of the developing mouse cortex. Extending the current knowledge about the diversity of this class of transcripts by the identification of nearly 4,000 new circRNAs, our study is the first to provide the full sequence information and expression patterns of circRNAs in cell types representing the lineage of neurogenic commitment. We further exploited our data by evaluating the coding potential, evolutionary conservation, and biogenesis of circRNAs that we found to arise from a specific sub-class of linear mRNAs. Our study provides the arising field of circRNA biology with a powerful new resource to address the complexity and potential biological significance of this new class of transcripts.

## Introduction

In the last few decades, the field of RNA biology has witnessed impressive developments. Fuelled by new sequencing technologies, these included the comprehensive annotation of micro- and long noncoding (lnc) RNAs in various organisms and tissues, the characterization of RNA modifications and the new field of *epitranscriptomic* and the discovery of an entirely new class of noncoding RNAs: circular (circ) RNAs (Kosik, 2013).

CircRNAs are transcripts whose 3′ and 5′ ends are covalently linked in a nonlinear manner resulting in a so-called backsplice junction (Lasda & Parker, 2014; Vicens & Westhof, 2014). The lack of a 3′ poly(A) tail and 5′ capping provides this class of RNAs resistance to exonuclease activity and, thus, an average longer half-live as compared with linear RNAs (Suzuki et al, 2006; Vincent & Deutscher, 2006; Jeck et al, 2013). Transcripts with these characteristics have long been known, but until recently, circRNAs were primarily found in viruses (Kos et al, 1986), and although some reports indicated their origin also from eukaryotic genomes (Nigro et al, 1991; Capel et al, 1993; Zaphiropoulos, 1996), these were still considered a rarity or a byproduct of splicing with no specific function. This view was completely changed very recently after the identification of thousands of circRNAs, including some with regulatory functions during brain development (Salzman et al, 2012; Hansen et al, 2013; Jeck et al, 2013; Memczak et al, 2013; Piwecka et al, 2017).

Despite their abundance, predicting circRNAs remains burdensome and typically relies on bioinformatic tools identifying sequences across backsplice junctions from RNA sequencing data obtained upon depletion of ribosomal RNA (Szabo & Salzman, 2016). Although this has resulted in the prediction of thousands of potential circRNAs in cell lines or whole organs of many species (Salzman et al, 2012; Jeck et al, 2013; Memczak et al, 2013; Westholm et al, 2014; Filippenkov et al, 2015; Jakobi et al, 2016; Kristensen et al, 2018), this approach based on ribosomal RNA depletion has the critical limitation that reads that do not map on a backsplice junction cannot be assigned to a circular, as opposed to a linear, transcript, resulting in the exclusion of the overwhelming majority of the sequencing data. In turn, this makes it impossible to reliably reconstruct neither the full sequence nor the expression level of large pools of circRNAs. Overcoming these limitations, the use of the exonuclease RNase R triggers the digestion of linear RNAs, thereby allowing the isolation of circRNAs (Suzuki et al, 2006). To date, this strategy was applied to a few cell lines or whole organs (Jeck et al, 2013; Ashwal-Fluss et al, 2014; Bachmayr-Heyda et al, 2015; Jakobi et al, 2016; Yang et al, 2017), but the expression patterns of circRNAs in specific cell types in physiological conditions is not known, and a comprehensive reconstruction of their sequence is yet to be reported.

---

[1]CRTD-Center for Regenerative Therapies Dresden, School of Medicine, Technische Universität Dresden, Dresden, Germany   [2]DRESDEN-concept Genome Center c/o Center for Molecular and Cellular Bioengineering, Technische Universität Dresden, Dresden, Germany

Correspondence: federico.calegari@tu-dresden.de
Martina Dori's present address is Center for Genome Research, Department of Life Sciences, University of Modena and Reggio Emilia, Modena, Italy

In addition to a poor classification, barely a handful of circRNAs have been suggested to be functionally relevant. For example, and beside their use as biomarkers in various diseases from cancer to diabetes (Bahn et al, 2015; Memczak et al, 2015; Abu & Jamal, 2016; Kulcheski et al, 2016), a study concluded that at least some circRNA, such as circ-ZNF609, may retain coding potential (Legnini et al, 2017). In addition, Drosophila's circMbl was found to interact with MBL protein resulting from the linear form of the same transcript (Ashwal-Fluss et al, 2014). Finally, and perhaps as the only circRNA-mediated molecular mechanism underlying a specific cellular effect, Cdr1as/ciRS-7 was shown to act as a sponge for miR-7 during development (Hansen et al, 2013; Memczak et al, 2013), and its depletion altered synaptic transmission in adulthood (Piwecka et al, 2017).

From these and other studies, it emerged that the mammalian brain is the organ most enriched in circRNAs (Rybak-Wolf et al, 2015; Veno et al, 2015). Hence, given the scarce knowledge about circRNA function and lack of studies reporting their sequence and expression in specific cell types, we here decided to exploit a double reporter mouse line previously characterized by our group and allowing the isolation of proliferating neural stem cells, neurogenic progenitors, and newborn neurons based on the combinatorial expression of RFP and GFP, respectively (Aprea et al, 2013).

Specifically, during cortical development, proliferative progenitors (PPs) progressively switch from divisions that expand their population to divisions that generate more committed, differentiative progenitors (DPs), which in turn are consumed to generate newborn neurons (N) (Lui et al, 2011; Taverna et al, 2014). Hence, to identify the three coexisting subpopulations of PP, DP, and N during mouse corticogenesis, our group has generated a double reporter mouse line expressing (i) RFP under the control of the *Btg2* promoter and identifying the switch of PP to DP and (ii) GFP under the control of the *Tubb3* promoter as a marker of newborn N (Aprea et al, 2013). Validating this approach, the combinatorial expression of the two reporters allowed the isolation of PP (RFP–/GFP–) from DP (RFP+/GFP–) and N (GFP+, irrespective of RFP) and identification and validation of transcription factors, lncRNAs, and epigenetic modifications functionally involved in neurogenic commitment (Aprea et al, 2013, 2015; Artegiani et al, 2015; Noack et al, 2019). Hence, we here decided to further exploit this mouse line and provide the arising field of circRNA biology with the first resource identifying circRNAs sequence and expression patterns in specific cell types of the developing mammalian cortex.

## Results

### Comprehensive identification of cell type–specific circRNAs of the mouse cortex

To identify cell type–specific circRNAs, we FAC-sorted PP, DP, and N, each in three biological replicates, from the mouse cortex at embryonic day (E) 14.5 as previously described (Aprea et al, 2013). Total RNA was then treated with RNase R to degrade linear RNAs, which we found to be very efficient in reducing the levels of linear transcript down to undetectable levels even among those with the highest expression levels and most stable predicted secondary structure (Fig S1A). This was then followed by 150-bp single-end,

strand-specific, high-throughput sequencing. The reads were then aligned to the reference mouse genome (mm9) and unmapped reads used to identify the circularizing, backsplice junctions predictive of putative circRNAs (Fig S1B).

As a first step to assess the sequence of circRNAs and their expression in cortical cell types, we collected their genomic co-ordinates putting together all replicates of PP, DP, and N and obtaining an initial set of 6,033 putative circRNAs (Fig 1A). Then, according to their genomic locations, we selected genic transcripts whose start and end sites coincided with the annotated start and end site of an exon and separated them from the remaining transcripts that included genic transcripts with ends not coinciding with exons, antisense or intergenic ones (Fig S1D). Within the former, we separated sequences belonging to introns from exons, whereas the latter were considered as a single exon. Finally, we calculated the relative reads per kilo base per million (RPKM) value of each intron and exon (Fig 1A and see the Materials and Methods section).

Given the need to establish an unbiased minimum threshold of RPKM to define "*expression*," we next selected 10 predicted genic circRNAs from our dataset, including six that were not described by previous studies. We then cloned and sequenced these circRNAs from RNase R–treated lysates from the E14.5 mouse lateral cortex and found that in all cases, the predicted exon(s) (2–15 for each circRNA; 45 in total) were included in their sequence, whereas not a single intron (35 in total) could be detected (Fig 1B). Hence, we chose the highest RPKM value previously calculated among these predicted, but not detected, introns as a minimum threshold to define expression (RPKM > 3.5). As a result, this allowed us to re-define among the original list of 6,033 putative circRNAs the 5,073 fulfilling our criteria for expression (Supplemental Data 1). Furthermore, we selected form this list six genic circRNAs for which introns were predicted as part of their sequence and designed divergent primers to validate their existence. Again, while confirming the presence of exons in these circRNAs, we were unable to confirm intronic sequences (Fig S1C), leading us to conclude that the genic circRNAs in our list are primarily exonic and that reads mapping on intronic locations may derive from lariats. As a result, we decided to exclude intronic sequences from any subsequent analyses (these sequences were, however, left in the list of expressed features provided in Supplemental Data 1).

Next, as a validation of our approach, we selected 30 circRNAs among this list of 5,073, including nine belonging to the bottom 30% and four to the bottom 10% in expression levels. Subsequent real time PCRs (RT-PCRs) were performed from the E14.5 mouse cortex, with or without RNase R treatment, using either divergent primers spanning over the backsplice junctions (Fig S1E) or convergent primers for two linear transcripts used as internal negative control of the enzymatic treatment (GAPDH and Ezh2_linear). This confirmed the presence of the vast majority (80%) of the selected circRNAs (Fig 1C). Importantly, failure to detect some circRNAs was independent from their predicted expression levels pointing out more a suboptimal choice of primers than false positives in our analysis. From here, we then reconstructed the comprehensive list, sequence, and expression levels of the 5,073 circRNAs detected in PP, DP, and N and representing the neurogenic lineage during corticogenesis that we named **CiCo** for **Ci**rcRNAs of the mouse **Co**rtex (Fig 1A and Supplemental Data 1).

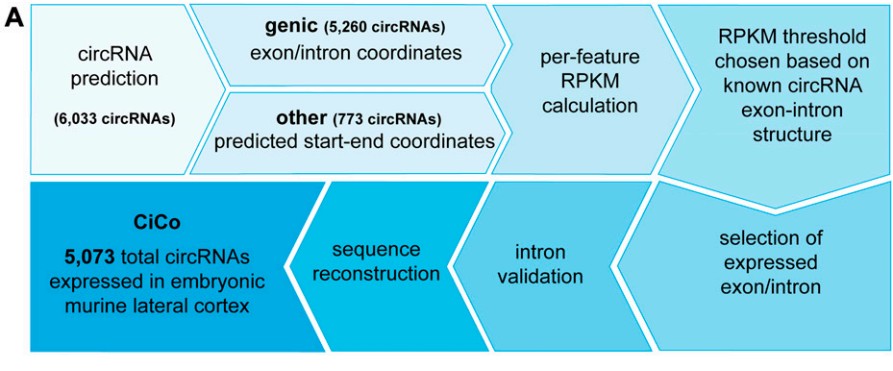

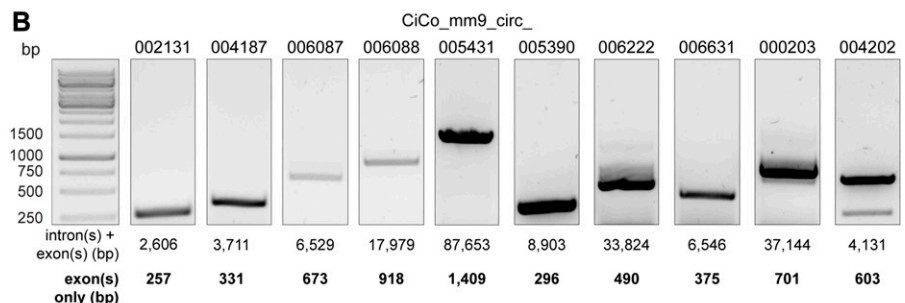

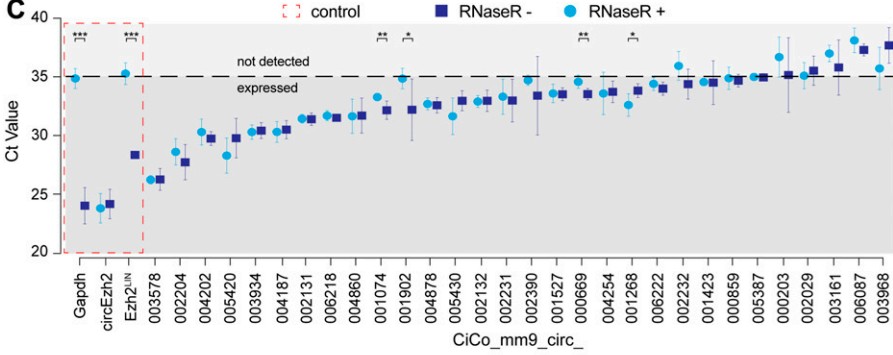

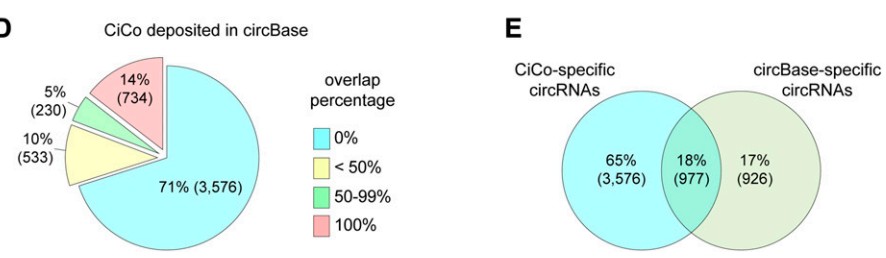

**Figure 1. circRNA prediction, sequence determination, and validation.**
**(A)** Strategy for the identification of circRNAs and determination of their sequence and expression in cortical cell types. **(B)** DNA gel electrophoresis upon PCR amplification of 10 circRNAs identified as in (A). Full gels are available in Fig S2. Note the detection of bands with a molecular weight consistent with the expected exonic, but not intronic, regions. **(C)** Graph representing the cycle threshold (Ct) values upon RT-PCR of total RNA from the E14.5 mouse cortex, with or without a prior treatment with RNase R (as indicated) and using divergent primers for 30 predicted circRNAs together with linear and circular controls (red dashed line). Dashed black line indicates the Ct threshold of detection. N = 2, n = 3; bars = SDs (*P < 0.05; **P < 0.01; ***P < 0.001; t test). **(D, E)** Distribution of CiCo's entries overlapping circRNAs deposited in circBase and (E) Venn diagram representing the proportion of circBase- and CiCo-specific circRNAs as well as common ones.

We next compared CiCo with the most complete resource of circRNAs available: circBase (Glazar et al, 2014). In particular, because of the stage- and tissue-specific expression of circRNAs, we selected from circBase murine circRNAs predicted from whole embryos and brains. Considering total or partial overlaps, down to the single nucleotide, as known circRNAs we found that our dataset extended this list by nearly three-fold adding to the ~1,900 transcripts of circBase ~3,700 new circRNAs of CiCo (Fig 1D) for a total of ~5,000 unique circRNAs (Fig 1E). As previously shown in the case of lncRNAs (Aprea et al, 2013, 2015), this high rate of novel circRNAs found in our study relative to previous reports highlights the power of cell type–specific analyses, allowing the identification of transcripts enriched in specific cell populations and being diluted out when considering bulk tissues or whole organs. In turn, this suggests a new layer of complexity in circRNA biology as these appeared to be not only stage- and tissue specific but also within the same developmental stage and tissue, to retain highly specific expression in individual cell types, which we sought to dissect next.

## General features and differential expression of CiCo

Previous studies reported that ~80% of the predicted circRNAs overlap in the sense strand with genes (Jeck et al, 2013; Memczak et al, 2013; Salzman et al, 2012). In our dataset, 97% of the expressed

sequences overlapped genes with the remaining 3%, including either antisense or intergenic circRNAs (Fig 2A). This seemed in contrast with the similar proportion of sense and antisense transcripts among lncRNAs (Aprea & Calegari, 2015), which can be explained by the overall lower expression of this class of transcripts biasing against the detection of circRNAs derived from them. Regarding length distribution and exon density, we found that most circRNAs (90%) were less than 1 kb long, primarily 250–500 bp, and including on average 2–3 exons (Fig 2B). As expected, a linear correlation was found between the length of circRNAs and the number of exons that they included, with no specific bias in their distribution across or within chromosomes (data not shown).

Next, we analysed CiCo's expression profiles of PP, DP, and N. In particular, we focused on the differentially expressed circRNAs considering a 50% threshold (i.e., a fold change [FC] by ≥1.5 or ≤0.67 for up- or down-regulation, respectively) (Fig 2C). Whereas a substantial proportion (42%) of circRNAs showed no significant change in expression among cell types, subdividing differentially expressed circRNAs into the possible patterns of up- or down-regulation during the neurogenic lineage revealed a distribution that was remarkably similar to that found in linear transcripts (Aprea et al, 2013) (Fig 2C). In addition, when analysing the biological functions and processes of the parental genes enriched in each group, we found that circRNAs up-regulated during neurogenesis were primarily associated with synaptogenesis and neuronal development. Conversely, down-regulated circRNAs were associated to cell cycle and regulation of transcription (data not shown).

Among transcripts showing specific expression patterns, an intriguing group of circRNAs emerged that was transiently up- or down-regulated specifically in DP compared with both PP and N (65 and 183, respectively) (Fig 2C). As previously shown by our group (Aprea et al, 2013), these patterns of expression account for a small, underrepresented proportion of differentially expressed transcripts (1–2%). Yet, at least among mRNAs and lncRNAs, many from this small subset of genes revealed to play key roles in corticogenesis (Aprea et al, 2013; Artegiani et al, 2015), raising the possibility that this also applies to circRNAs here identified for the first time.

## Properties of circRNAs: miRNA-sponging potential, translation, and evolutionary conservation

Our novel assessment of the sequence of circRNAs gave us the possibility to gain new insights into their putative function(s). Since sponging of miRNAs by Cdr1as was the first suggested role for a circRNA (Hansen et al, 2013; Memczak et al, 2013), we searched within CiCo for miRNA seed sequences. When assessing the highest possible number of seed sequences present in any given circRNA that would potentially target any given miRNA, we found that the highest number was reached by Cdr1as (CiCo_mm9_circ_006933) for which 99 seeds specific for miR-7b and, among others, a single seed for miR-671 were predicted as previously described (Memczak et al, 2013). Despite this extreme example, only a handful of circRNAs displayed a number of seeds for a single miRNA that exceeded five (Fig 3A). Moreover, since this analysis did not account for the potential of circRNAs to sponge more miRNAs independently, we expanded our search to include all possible miRNA binding to a single circRNA with no regard to the number of seeds. With this approach, we identified in CiCo_mm9_circ_006344 the circRNA with the highest

number of potentially binding miRNAs for a total of nearly 1,400 different miRNAs of which four (miR-1187, 466i, 466k, and 669c) with a high number of seeds (46, 43, 37, and 30 respectively; Fig 3A and Supplemental Data 1). However, such a high promiscuity in miRNA-binding sites and limitations in the bioinformatic tools to predict sponging properties makes it difficult to infer the biological significance of this circRNA.

In addition to miRNA sponging, a recent study has suggested that at least one circRNA, circ-ZNF609, is translated into a peptide (Legnini et al, 2017). Hence, we searched for possible open reading frames (ORFs) among all circRNAs and found that a remarkably high proportion (nearly 4,000; i.e., >70%) contained at least one putative ORFs with ≥150 nt in length (Fig 3B, left). To increase our confidence in this result, we assessed the Codon Adaptation Index (CAI) measuring the relative codon usage as a function of gene expression (Sharp & Li, 1987). To this end, we used CAIcal (Puigbo et al, 2008) to compute the index of each predicted ORF within circRNAs and referred this to the expected value calculated from a pool of 500 shuffled sequences reflecting the whole ORF population (eCAI = 0.744) that was used as a significance threshold. We found that 72% of all predicted ORFs and belonging to ~50% of all circRNAs had an index higher than expected (Fig 3B, right), raising the possibility that these are potentially translated. However, 97% of CiCo's sequences derive from annotated genes, implying that this abundance of ORFs might simply reflect their sharing of some of, if perhaps not all, the features of coding genes. To assess this, we repeated our analyses (ORF prediction and CAI calculation), but this time considering random sequences generated not only from the entire genome but also from the coding transcriptome as two independent negative controls. Although we could find a comparable number of putative ORFs from CiCo and our two negative control datasets (5,966, 3,239, and 4,372, respectively), only 52% of all random genomic ORFs and 66% of the transcriptomic ones were found to have a CAI value above the eCAI thresholds. Remarkably, when comparing the fraction of significant and nonsignificant ORFs within CiCo with the two control datasets, we found highly significant differences with both (chi-squared test, two-tailed $P$-value < 0.001) (Fig 3B, right) suggesting that the coding potential of CiCo is not only an inherited feature resulting from their origin from coding genes but might potentially account for their function.

We next assessed the evolutionary conservation of CiCo. To this end, we considered not only the sequence of the mature circRNAs but also their flanking regions (200 bp up- and down-stream) as these might be important for their biogenesis (Jeck et al, 2013; Ashwal-Fluss et al, 2014). Again to obtain an appropriate reference as negative control, we generated for each circRNA and flanking regions a random sequence of the same size and reflecting the features of such circRNA. Specifically, we used any random genomic region for intergenic circRNAs or transcriptome-specific and intronic sequences for genic circRNAs and their flanking regions, respectively. We next computed the average conservation score of all these regions, making use of the phyloP score (a per-base value obtained by the alignment of murine genome against 30 other vertebrates, UCSC Genome Browser [Casper et al, 2018]) and compared it with the one calculated for the respective random sequences. We found that genic circRNAs were more conserved than their reference random sequences and that despite an average lower conservation score, their flanking regions were also significantly more conserved (Fig 3C; top). As

**A**

### CiCo annotation

3% (110)

97%
(4,963)

■ genic

■ other ── genic
         ── antisense
         ── intergenic

**Figure 2.   General features of CiCo.**
**(A)** Number and proportion of CiCo and their genomic features showing they almost entirely (97%) derive from genic regions. **(B)** Length (left) and exon number (for genic transcripts, right) distribution of circRNAs. **(C)** Differential expression (50% i.e.: FC ≥1.5 or ≤0.67; no FDR being applied) of CiCo expressed in PP (grey), DP (red), and N (green). Abundance and proportion of circRNAs detected in each cell type are indicated and represented proportionally to the area of circles and pattern of expression.

**B**

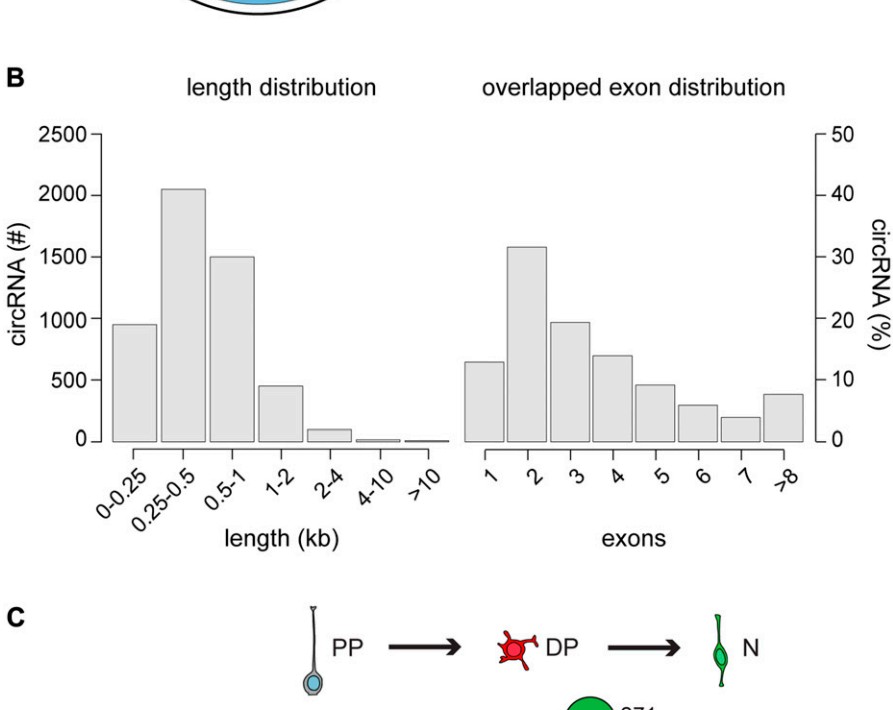

length distribution                    overlapped exon distribution

**C**

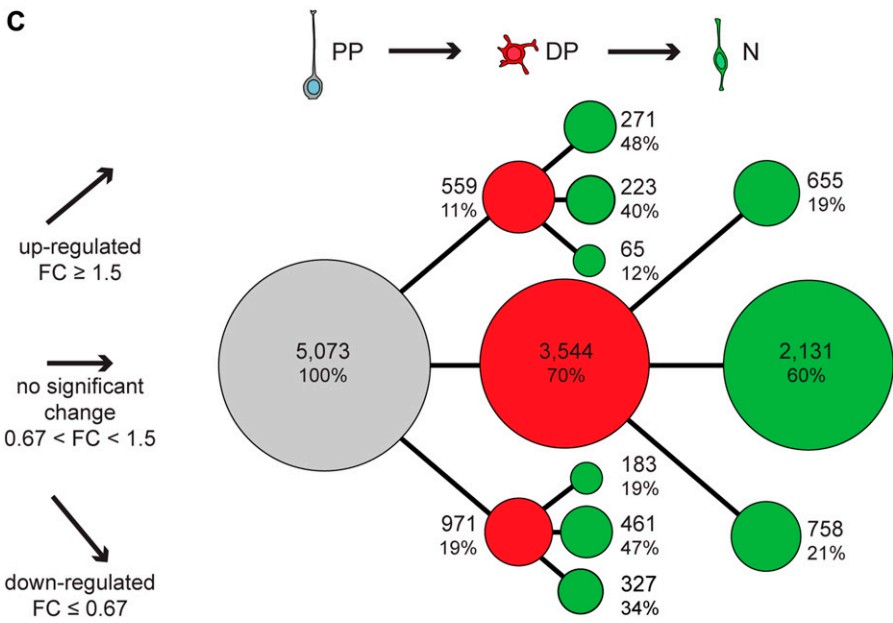

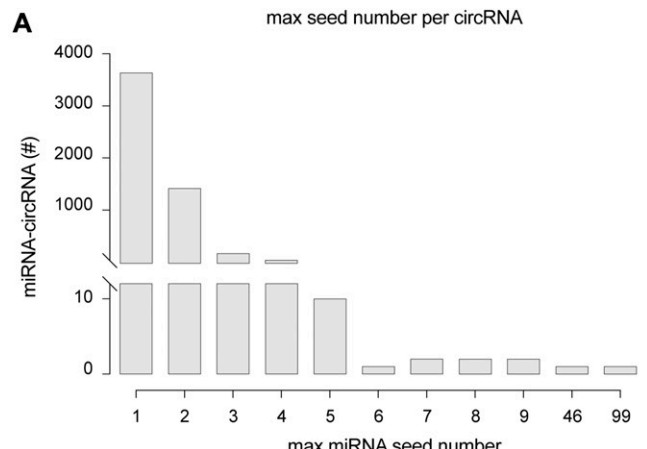

max seed number per circRNA

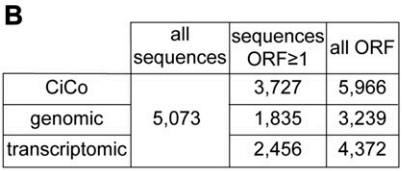

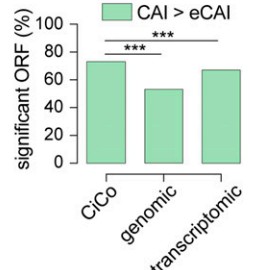

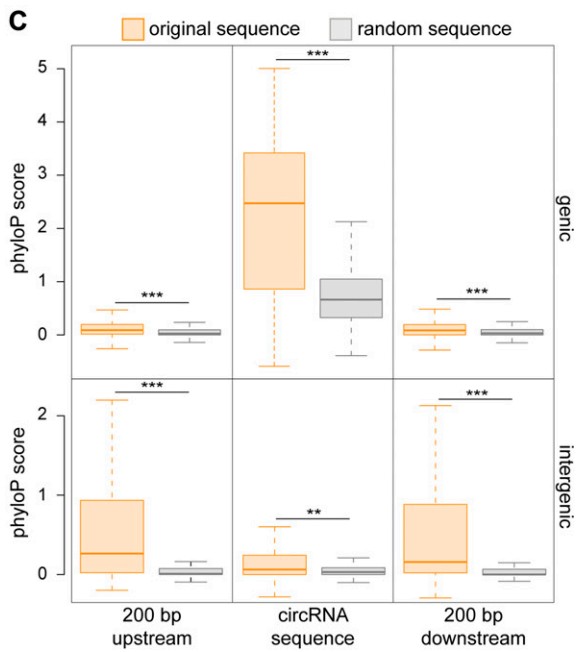

**Figure 3. Seed sequences, coding potential, and evolutionary conservation of circRNAs.**
**(A)** Distribution of the maximum number of seed sequences for any miRNA found across CiCo. Note that the overwhelming majority of circRNAs only displayed few (1–4) seeds. **(B)** Total ORFs (left) and proportion of those with a CAI value above expected (right) found in CiCo or random control datasets (***$P < 0.001$; chi-squared test). **(C)** Whiskers–box plots showing the distribution of phyloP scores for CiCo sequences (orange) and relative shuffled controls (grey) (**$P < 0.01$; ***$P < 0.001$; $t$ test).

this again can be influenced by circRNA origin from expressed genes, we next assessed the conservation of the subgroup of intergenic circRNAs that, by definition, are not overlapping any annotated gene. Strikingly, both the circRNAs themselves as well as their 200-bp up- and down-stream regions were also significantly more conserved than their random counterparts (Fig 3C; bottom). Hence, similarly to their coding potential, the conservation of circRNAs seems not only an inherited feature resulting from their sharing of sequences with expressed genes but may actually reflect their function.

**CircRNA biogenesis is independent from alternative splicing**

Finally, because nearly all (97%) CiCo transcripts are overlapping genes, we asked if the abundance of circRNAs correlated with that of their respective linear mRNA, which could reveal the mechanisms underlying their biogenesis. In this context, we sought possible mechanisms by which biogenesis of circRNAs may be controlled. We speculated that this may occur at the level of transcription by the synthesis of different pre-RNAs that ultimately mature into circular, rather than linear, transcripts. Alternatively, a circular or linear RNA may result by splicing of a single pre-RNA precursor. Although not mutually exclusive, the two mechanisms may occur either un-specifically without any regulation at the level of individual transcripts or cell types or, alternatively, only apply to a specific subclass of genes. Ultimately, distinguishing between these possibilities may help understanding the biological significance of circRNAs.

To this end, we took advantage of the previous poly(A) transcriptome assessment of PP, DP, and N reported by our group (Aprea et al, 2013, 2015) and compared the expression levels of each circRNA and its corresponding linear counterpart in each cell type. We found a strong positive correlation between the expression of mRNAs and circRNAs (Pearson's score ~0.8 in all cell populations) (Fig 4A), implying that an increase in a circRNA did not result in a decrease in its linear form. This in turn led us to reject the hypothesis of regulation at the level of transcription of different pre-RNAs. Next, if synthesis of a circular versus linear RNA should be regulated by splicing from a common transcript, then an increase in such circRNA should inversely correlate with the usage of the exon(s) in common with the linear transcript. To assess this, we performed differential exon usage analyses of PP, DP, and N (to be described elsewhere) and assessed whether exon(s) used in common by both a circular and linear transcript would display an inverse FC from one cell population to the following (e.g., from PP to DP and from DP to N). Surprisingly, barely 0.5% of all circRNA–exon pairs showed an opposite FC pattern with regard to their linear counterparts, whereas the vast majority shared the same pattern of either up- or down-regulation (Fig 4B). In turn, this questioned a mechanistic link between mRNA splicing per se and circRNA biogenesis. To address this, we next selected all genes resulting in the expression of at least two linear isoforms (4,146 genes out of 19,802 expressed) resulting from alternative splicing as one specific mechanism of splicing and investigated what proportion of circular RNA was produced among these genes. We found that only 49% of circRNAs were generated from alternatively spliced genes and that among these, only ~0.2% shared the very same exon(s) in all three cell populations with the differentially spliced linear transcript (Fig 4C).

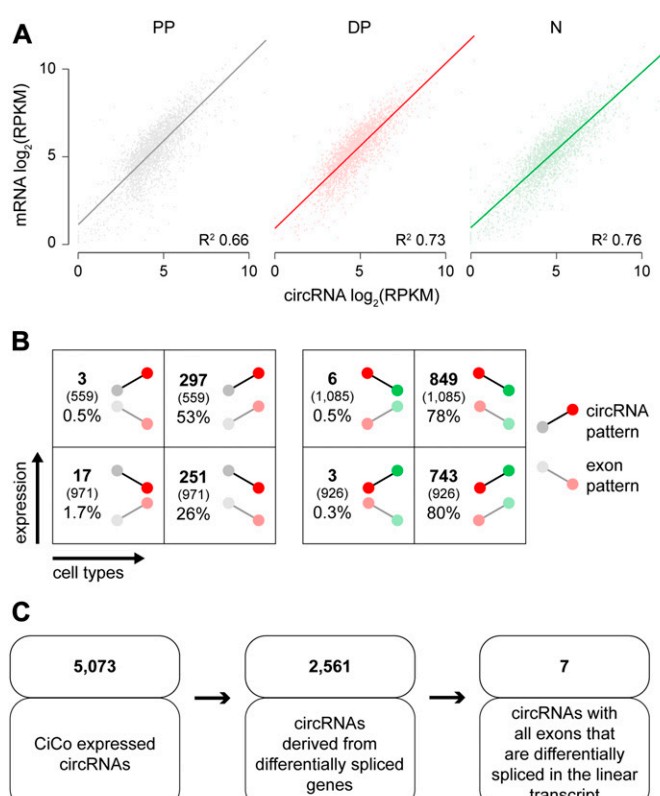

**Figure 4. CiCo relation to linear transcripts.**
**(A)** Scatterplot log$_2$(RPKM) of CiCo and mRNA counterparts in cell types (colours as in Fig 2C). R$^2$ for each regression line is indicated. **(B)** Number and proportion of circRNAs having either an opposite or the same FC pattern as the exon of the linear counterpart (50% i.e.,: FC ≥1.5 or ≤0.67; no FDR being applied; colours as in Fig 2C). **(C)** Number of expressed circRNAs that result from differentially spliced genes during the neurogenic lineage.

Taken together, our data indicate that the overall expression of circRNAs correlates with that of their linear RNAs but is not a byproduct limited to transcripts and exons undergoing alternative splicing and, hence, can be regulated independently by mechanisms that are still unknown.

# Discussion

Here, we provided the arising field of circRNA biology with the first resource describing their sequence and cell-specific expression during mammalian cortical development. This has allowed us to increase by threefold the number of circRNAs currently known, pointing out their full diversity and specificity in cell populations representing the lineage to neurogenic commitment. More importantly, our work has allowed us to reveal novel features of these elusive transcripts that are important to infer their significance and that could not be deduced based on the previous knowledge of backsplice junctions alone.

We found that genic circRNAs are primarily composed by exons of genes encoding for mRNAs or lncRNAs and that a strong correlation existed in the levels of expression of circular and linear

pairs of transcripts. Our data were inconsistent with both the synthesis of circular-specific, pre-RNAs during transcription as well as the involvement of alternative splicing as a mechanism underlying their biogenesis. Although more studies are needed to address these aspects, certain features emerging from our study seemed to support the notion that circRNAs may be a generic byproduct of splicing, whereas others highlighted their specificity both in terms of biogenesis and putative biological significance. To start with, CiCo transcripts were found to derive only from a specific group of linear transcripts and consistently included certain exons but not others. In addition, both the coding potential and evolutionary conservation of CiCo revealed to be much higher than expected by chance even after accounting for their origin from exonic regions.

Although it is clear that a significant proportion of circRNAs might nonetheless lack a function despite these features, our study provides the field with a new resource for the identification of biologically relevant circRNAs, fuelling future studies on their molecular regulation and role.

# Materials and Methods

### Animal care, cell sorting, and RNA extraction

Animal experiments were approved by the Landesdirektion Sachsen (24-9168.11-1/41 and TVV 39/2015) and carried out in accordance with the relevant guidelines and regulation. Pregnant *Btg2*$^{RFP}$/*Tubb3*$^{GFP}$ double heterozygous mice were anaesthetised with isoflurane and euthanized through cervical dislocation. Brains from E14.5 embryos were collected and the lateral cortex isolated after removal of the meninges and ganglionic eminences. The Neural Tissue Dissociation kit with papain (Miltenyi Biotech) was used according to the manufacturer's protocol to obtain a cell suspension for FAC-sorting or RNA extraction. The cells were resuspended in ice-cold PBS and supplied with 10 µl of 7-AAD to assess cell viability (BD Pharmigen) and sorted with a BD Aria III FACS as previously described (Aprea et al, 2013). Sorted cells were collected in PBS, centrifuged (600 g, 5 min), and RNA extracted using Quick RNA Mini Prep (Zymo Research) according to the manufacturer's protocol.

### CircRNA sequencing, annotation, and validation

Total RNA was denatured for 3 min at 70°C, then treated with RNase R (Epicentre) for 1 h at 40°C, and finally supplied with DNase I (Invitrogen) for 15 min at room temperature. The reaction was then cleaned with RNA Clean & Concentrator (Zymo Research) and cDNA libraries prepared using the NEB Next Ultra Directional RNA Library Prep kit without mRNA enrichment. Effectiveness of RNase R treatment was assessed by quantitative real time PCR of six transcripts with the highest free energy per nucleotide in their most stable predicted secondary structure according to RNAfold (-p −noLP −temp = 40; primers in Table S1C). Samples were sequenced on an Illumina HiSeq 2500 with a read length of 150 bp and resulting reads were aligned using gsnap (Wu & Nacu, 2010) with mm9 and ENSEMBL Genes (v67 [Flicek et al, 2013]) as the reference genome and trancriptome, respectively. Besides adapter removal, reads

were not trimmed, and overall quality was evaluated by the percentages of mapped and unmapped reads that resulted in similar values to the only comparable study reported to date (Jeck et al, 2013) (Fig S1B). To predict putative circRNA, unmapped reads were retrieved and then analysed using the "find_circ" pipeline (version 1) with default parameters (Memczak et al, 2013). No filter was applied on the number of reads identifying the circularising junction. For the genomic location, predicted circRNAs were overlapped with mouse ENSEMBL Genes (v67 [Flicek et al, 2013]) and considered them genic when their start/end base coincided with the start/end of annotated exon(s). circRNA overlapping genes but with other start/end points were grouped together with intergenic and antisense ones (Fig 2A, termed as "other"). Overlap with circBase was carried out using bedtools (Quinlan & Hall, 2010) and intersecting predicted circRNAs with the .bed file relative to mouse (Glazar et al, 2014). Different levels of minimal reciprocal overlap were set within bedtools options to take into account also single-nucleotide overlap between data. For validation, 1 μg of total RNA from un-sorted lateral cortex was treated as described previously for sequencing, with one sample digested either with RNase R or water and then converted into cDNA using 200 U of reverse transcriptase and 50 ng of random hexamers according to the SuperScript III (Invitrogen) kit. The resulting cDNA was diluted either 1:10 for RNase R or 1:100 in case of water treatment and 1 μl used as a PCR template with divergent primers (designed with Primer3 ([Untergasser et al, 2012]) and spanning over the circularising junction. All primer pairs were tested with the In-Silico PCR tool from UCSC to minimize byproducts (Kent et al, 2002) (primers in Table S1A).

## Sequence prediction, differential expression, and conservation

First, exonic and intronic sequences of previously annotated genic circRNAs were separated. Next, we retrieved the coordinates of these exon(s) from ENSEMBL Genes (v67 [Flicek et al, 2013]), whereas intron coordinates were obtained through command line starting from the exonic ones. circRNAs annotated as intergenic, antisense, or genic with unusual start/end points were considered as a single exon. Exonic and intronic coordinates were kept separate, and featureCounts (Liao et al, 2014) was run using them as reference files to obtain a per-feature read count on which RPKM values were calculated as follows: $RPKM = \frac{read\_count}{R/L}$, where R = library size × $10^{-6}$ and L = circ length × $10^{-3}$. RPKM were averaged across biological replicates. Threshold for expression was set as the highest RPKM of the predicted, but not detected, introns within 10 validated circRNAs, with additional six circRNA used to validate the absence of introns from which we redefined the set of expressed features (primers in Table S1B and D). For the differential expression analysis, we run again featureCounts (Liao et al, 2014) using the set of expressed features as reference and specifying a meta-feature count to automatically sum the reads from different features of the same circRNA. The resulting table was analysed with DESeq2 (Love et al, 2014), considering FCs by ≥1.5 or ≤0.67 (no FDR applied). Clustering analysis was used to evaluate enriched terms using DAVID (Huang da et al, 2009). Finally, circRNA conservation was assessed by computing the phyloP score for the expressed sequence as well as for 200 bp up- and down-stream of each circRNA. To compare our sequences, we generated a shuffled version with the same sizes using bedtools (shuffleBed) and accounting for their genomic location. In particular, for genic circRNAs, the shuffled sequences were obtained from a reference file that included exons and/or introns as appropriate and derived from ENSEMBL (v67). In the case of intergenic circRNAs (both sequence and flanking regions), the shuffled sequences were chosen from the entire genome. BEDOPS (Neph et al, 2012) was used to sort regions per genomic location and to split into a per-base coordinate file. BEDOPS was again used to retrieve the per-base phyloP score from the UCSC table containing the conservation score for mouse versus 30 vertebrate genomes (phyloP30wayAll). For shuffled sequences relative to genic circRNAs, we built a custom phyloP score table in which the values relative to all exons were included. We finally reconstructed the score per sequence by averaging the phyloP values for each circRNA. These steps were performed for all files (original and shuffled sequences).

## Prediction of miRNA seeds, ORFs, and exon usage correlation

Seed prediction was performed using miRanda (John et al, 2004) with default parameters and with input the FASTA file for the circRNA sequences obtained with bedtools (getfasta, with the expressed feature coordinates file as input) and miRBase-downloaded mouse mature miRNA sequences (v22) (Kozomara & Griffiths-Jones, 2014). The output generated was then parsed to a more manageable form, and the seed category was added according to Bartel (Bartel, 2009). To count the number of seed for each miRNA present on a circRNA, we first subset the parsed miRanda output by seed category (removed everything that did not have a recognized seed type), alignment score (≥150), and by free energy (ΔG ≤ −19) (Bartel, 2009), and then we counted and reported the number of unique miRNA-circRNA combination and sorted by number of occurrences (top to bottom). We predicted the presence of ORF using the standalone version of National Center for Biotechnology Information's ORF-finder (Sayers et al, 2011) using the FASTA sequences of the expressed circRNA as input. We restricted the search by requiring a minimum ORF length of 150 nt and setting the start codon as ATG only; we also ignored nested ORFs and searched only on the strand of the sequence itself. We then supplied the resulting sequences to the CAIcal web server (Puigbo et al, 2008) together with the codon usage for mouse (Nakamura et al, 2000). Expected CAI value (eCAI) was calculated several times using the Markov method with 95% of confidence and the final value obtained by averaging all the calculated ones. As controls, we generated two shuffled dataset using as reference either the entire genome or the transcriptome only. For this two control datasets, the same ORF prediction and CAI calculation were performed. Differences of our dataset with the two control ones were assessed through chi-squared test with one degree of freedom. Exon usage data were obtained by poly(A) enrichment and PE sequencing of our three cell populations (to be reported elsewhere). For the full mRNA, we selected the RPKM values for the linear corresponding to the circRNAs. For linear versus circRNA expression, we computed the average RPKM for each cell population and compared the $\log_2(RPKM)$, plotting their relative abundance and computing an overall Pearson's correlation score. In case of linear exon versus circRNA, the corresponding read counts were analysed with DESeq2 to compute the FC of the exon shared by circRNAs and linear transcript.

## Data availability

All custom scripts can be obtained upon request. Sequencing data generated during the present study are available at Gene Expression Omnibus repository (GSE117009).

# Supplementary Information

# Acknowledgements

We thank the facilities at Max Planck Institute of Molecular Cell Biology and Genetics (MPI-CBG) and Center for Regenerative Therapies Dresden (CRTD) for maintenance of mouse lines, sequencing, and FACS. We are grateful to the members of Silvio Bicciato's Lab (University of Modena) and Dr. Michael Hiller (MPI-CBG) for suggestions and discussion. We also thank Prof. Katja Nowick and Maria Beatriz Walter Costa for insights into RNA structure prediction. This work was funded by the CRTD, the School of Medicine of the TU-Dresden, the Italian Epigenomics Flagship Project (Epigen), the Deutsche Forschungsgemeinschaft (DFG) grant DFG CA 893/9-1, and a fellowship from the Dresden International Graduate School for Biomedicine and Bioengineering (DIGS-BB) awarded to M Dori.

## Author Contributions

M Dori: conceptualization, data curation, formal analysis, validation, investigation, methodology, and writing—original draft, review, and editing.
L Haj Abdullah Alieh: validation and investigation.
D Cavalli: validation and investigation.
S Massalini: validation and investigation.
M Lesche: data curation.
A Dahl: data curation.
F Calegari: conceptualization, supervision, funding acquisition, and writing—original draft, review, and editing.

## Conflict of Interest Statement

The authors declare that they have no conflict of interest.

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
