## [Reviewer comments · Life Science Alliance]

Life Science Alliance

Sequence and Expression Levels of Circular-RNAs in Progenitor Cell Types During Mouse Corticogenesis

Martina Dori, Leila Haj Abdullah Alieh, Daniel Cavalli, Simone Massalini, Mathias Lesche, Andreas Dahl, and Federico Calegari

DOI: <https://doi.org/10.26508/lsa.201900354>

Corresponding author(s): Federico Calegari, DFG-Research Center and Cluster of Excellence for Regenerative Therapies, Faculty of Medicine, Technische Universität Dresden and Martina Dori, University of Modena and Reggio Emilia

Review Timeline:

Submission Date:	2019-02-21
Editorial Decision:	2019-02-22
Revision Received:	2019-03-19
Editorial Decision:	2019-03-19
Revision Received:	2019-03-22
Accepted:	2019-03-22

Scientific Editor: Andrea Leibfried

Transaction Report:

Please note that the manuscript was previously reviewed at another journal and the reports were taken into account in the decision-making process at Life Science Alliance. Since the original reviews are not subject to Life Science Alliance's transparent review process policy, the reports cannot be published.

February 22, 2019

Re: Life Science Alliance manuscript #LSA-2019-00354

Prof. Federico Calegari
DFG-Research Center and Cluster of Excellence for Regenerative Therapies, Faculty of Medicine,
Technische Universität Dresden
Fetscherstr. 105
Dresden 01307
Germany

Dear Dr. Calegari,

Thank you for submitting your manuscript entitled "Assessment of Circular-RNAs in Progenitor Cell Types of the Developing Mammalian Cortex" to Life Science Alliance. The manuscript was assessed by expert reviewers at a different journal twice before, and you provided those reviewer reports to us.

One of the previous reviewers appreciated the changes introduced in revision and only expected you to further discuss sponging potential of circRNAs as well as sequence conservation of circRNAs between mice and human. The other reviewer, however, questioned the resource value of your findings and thinks that your sequencing reads may derive from linear molecules.

We therefore decided to seek arbitrating advice on your work in light of the existing reviewer reports. I am copying the expert's advice below. The advisor thinks that toning down your conclusions and a further minor revision to address the comments of reviewer #1 and the technical recommendations of reviewer #2 will make your manuscript suitable for publication in Life Science Alliance. We would thus like to invite you to provide such a revised manuscript.

The typical timeframe for revisions is three months.

Thank you for this interesting contribution to Life Science Alliance. We are looking forward to receiving your revised manuscript.

Sincerely,

B. MANUSCRIPT ORGANIZATION AND FORMATTING:

COMMENTS FROM THE ADVISOR:

The identification of circRNAs they do is perhaps not as stringent as possible (there are now some cleaner methods that don't just use RNase R, but combine it with A tailing + oligo dT selection etc.,

but these methods are very recent), but still most of the circRNAs they find are likely real, and the users of this resource will anyhow validate the circRNAs they will work further with. I think they should address the comments of reviewer #1, do the technical recommendations of reviewer #2, and tone down their statements in accordance to the comments of both reviewers.

March 19, 2019

RE: Life Science Alliance Manuscript #LSA-2019-00354R

Prof. Federico Calegari
DFG-Research Center and Cluster of Excellence for Regenerative Therapies, Faculty of Medicine,
Technische Universität Dresden
Fetscherstr. 105
dresden, Dresden 01307
Germany

Dear Dr. Calegari,

Thank you for submitting your revised manuscript entitled "Sequence and Expression Levels of Circular-RNAs in Progenitor Cell Types During Mouse Corticogenesis". We appreciate your point-by-point response to the concerns previously raised as well as the changes introduced in the manuscript, and we would be happy to publish your paper in Life Science Alliance pending final revisions necessary to meet our formatting guidelines:

- please include the S file legends and table S1 in the main manuscript docx file and upload the S figure files as individual files
- please note that figure 3 legend mentions panels A-B-B instead of A-B-C
- please note that there is a mis-match between the source data in Figure S2A and the PCR gel shown in figure 1 - please fix.

A. FINAL FILES:

- An editable version of the final text (.DOC or .DOCX) is needed for copyediting (no PDFs).
- High-resolution figure, supplementary figure and video files uploaded as individual files: See our detailed guidelines for preparing your production-ready images, <http://www.life-science-alliance.org/authors>
- Summary blurb (enter in submission system): A short text summarizing in a single sentence the

study (max. 200 characters including spaces). This text is used in conjunction with the titles of papers, hence should be informative and complementary to the title. It should describe the context and significance of the findings for a general readership; it should be written in the present tense and refer to the work in the third person. Author names should not be mentioned.

B. MANUSCRIPT ORGANIZATION AND FORMATTING:

Sincerely,

Andrea Leibfried, PhD
Executive Editor
Life Science Alliance
Meyrhofstr. 1
69117 Heidelberg, Germany
t +49 6221 8891 502
e a.leibfried@life-science-alliance.org
www.life-science-alliance.org

March 22, 2019

RE: Life Science Alliance Manuscript #LSA-2019-00354RR

Prof. Federico Calegari
DFG-Research Center and Cluster of Excellence for Regenerative Therapies, Faculty of Medicine,
Technische Universität Dresden
Fetscherstr. 105
dresden, Dresden 01307
Germany

Dear Dr. Calegari,

Thank you for submitting your Resource entitled "Sequence and Expression Levels of Circular-RNAs in Progenitor Cell Types During Mouse Corticogenesis". It is a pleasure to let you know that your manuscript is now accepted for publication in Life Science Alliance. Congratulations on this interesting work.

DISTRIBUTION OF MATERIALS:

Again, congratulations on a very nice paper. I hope you found the review process to be constructive and are pleased with how the manuscript was handled editorially. We look forward to future exciting submissions from your lab.